# Mercury in Soil and Forage Plants from Artisanal and Small-Scale Gold Mining in the Bombana Area, Indonesia

**DOI:** 10.3390/toxics8010015

**Published:** 2020-02-22

**Authors:** Masayuki Sakakibara, Koichiro Sera

**Affiliations:** 1Sekolah Tinggi Ilmu Kesehatan Makassar, Jl. Maccini Raya No. 197, Makassar 90231, South Sulawesi, Indonesia; 2Graduate School of Science and Engineering, Ehime University, 2-5 Bunkyo-cho, Matsuyama, Ehime Prefecture 790-8577, Japan; 3Faculty of Collaborative Regional Innovation, Ehime University, 3 Bunkyo-cho, Matsuyama, Ehime Prefecture 790-8577, Japan; 4Cyclotron Research Centre, Iwate Medical University, 348-58 Tomegamori, Takizawa, Iwate 020-0173, Japan; ksera@iwate-med.ac.jp

**Keywords:** mercury pollution, soil, forage plants, ASGM

## Abstract

Mercury (Hg) contamination in soil and forage plants is toxic to ecosystems, and artisanal and small-scale gold mining (ASGM) is the main source of such pollution in the Bombana area of Indonesia. Hg contamination in soil and forage plants was investigated by particle-induced X-ray emission analysis of samples collected from three savannah areas (i.e., ASGM, commercial mining, and control areas) in the Bombana area. Hg contents of forage plants in the ASGM area (mean 9.90 ± 14 µg/g) exceeded those in the control area (2.70 ± 14 µg/g). Soil Hg contents (mean 390 ± 860 µg/g) were also higher than those in the control area (mean 7.40 ± 9.90 µg/g), with levels exceeding international regulatory limits. The Hg contents of 69% of soil and 78% of forage-plant samples exceeded critical toxicological limits. Thus, the Hg levels observed in this study indicate that contamination extending over large areas may cause major environmental problems.

## 1. Introduction

The United Nations Environment Programme (UNEP) has determined that Indonesia is the third-largest producer of mercury (Hg) emissions in the world, after China and India [1], with artisanal and small-scale gold mining (ASGM) being a major and increasing contributor [2,3]. With the increasing price of gold, ASGM is now undertaken in 180,000 locations in Indonesia [4], with the industry composed of more than a million workers in 27 provinces [2], generating an annual income of USD5 billion and representing ~7% of total gold production [5]. There are ~300,000 artisanal gold miners working in ~1000 areas throughout Indonesia [5], with the increasing workforce being due to shifts in employment from the agricultural and fishing sectors. Illegal gold mines in rural areas provide an alternative livelihood even though significant capital investment is required [4]. Savannah areas in Indonesia contain placer-type gold deposits, and the potential of ASGM attracts migration, creating social conflict and economic complexity [6].

Natural sources of Hg in the atmosphere include volcanic emissions (80–600 tonnes yr^–1^) [7], geothermal activity, cinnabar deposits, and forest fires, which cycle Hg through the atmosphere via dry and wet deposition, with >90% of released Hg entering the terrestrial environment [8]. 

ASGM is now the main source of Hg pollution globally [9], with 10–19 million workers in 70 countries using Hg to purify gold. The amount of Hg released is proportional to 5 units the volume consumed per unit of gold produced [10]. Hg pollution also arises from the nonferrous metal, coal, and cement industries, and mine-tailing waste (Figure 1). Discharge of Hg to the atmosphere results in contamination of soil, water, and vegetation before re-release through volatilization (‘latent emission’) [11], with soil chemistry and groundwater characteristics directly affecting the distribution and concentration of contamination [12]. Environmental risks are determined by the bioavailability of Hg [13]. Where degradation of methyl mercury is inhibited in the soil matrix [13], toxicity to forage plants and grazing animals results [14,15].

Previous studies [16,17,18] of the Hg contents of plants growing near various sources of Hg pollution, including gold mines, have revealed that contamination levels in plants growing around gold mines in various regions in Indonesia having environmentally toxic potential. These mine sites include Poboya, Palu city (6.5 mg Hg kg^–1^) [18]; Sumalata District, North Gorontalo Regency (up to 1.31 mg kg^–1^) [17]; and Buru Island, Maluku (0.08 mg kg^–1^) [16]. Plant samples from the Kaili coal mining region, China, and the Idrija mercury mine area, Slovenia, have high average Hg contents of 0.88 and 12.7 mg kg^–1^, respectively [11].

A new gold-mining site in the Bombana area, Southeast Sulawesi Province, Indonesia, has expanded over the last ten years, with potential Hg contamination of forage-plant animal feedstock. Gold processing activities there produce Hg-bearing particulates that disperse over large areas in the wind and are deposited in rain. Previous studies [6,19] of the area indicate widespread dispersion of Hg, resulting in elevated Hg concentrations in mining workers and cattle. There have been few studies of the effects of Hg pollution on soil and forage plants in savannah landscapes. To address this lack of information, this study focused on the distribution of soil Hg contamination around ASGM sites in the Bombana area and its effects on forage plants, with the aim of evaluating the environmental implications of ASGM.

## 2. Materials and Methods 

### 2.1. Study Area

Samples of soil and forage plants were collected during March 2016 to July 2017 from several locations (Figure 2) in the Rarowatu and North Rarowatu districts of Bombana, covering an area of 130 km^2^. The Rarowatu district includes ASGM and commercial mining areas, whereas the North Rarowatu district has no gold mining activity and is along the highway that is used here as a control area [19]. 

### 2.2. Sample Collection

#### 2.2.1. Soils 

Soil samples were collected from three principal locations: ASGM, commercial mining, and control areas [6]. Topsoil samples (0–30 cm) were collected from 26 areas (Figure 2) using standard sampling procedures [20]. Soil samples were also collected from around trommel machines at six commercial mining facilities, with a sampling depth of 0–30 cm and at a maximum distance of 3 m from the machine (in trommel houses, ore is ground with Hg during amalgamation, producing Hg droplets that are easily washed away and transported across large areas). Sample homogeneity and representativeness were ensured by sampling in a zig-zag pattern at three locations at each site, with locations recorded by GPS [21,22]. Samples were initially stored in plastic bags and later transferred to autoclavable polypropylene bags pending analysis.

#### 2.2.2. Forage Plants

Fresh forage-plants samples were collected from 20 areas and included cattle-feed grasses (fodder) from various farms. Samples are preserved in a herbarium for identification purposes [23]. At each sampling point, three types of forage plant were collected: *Imperata cylindrica*, *Megathyrsus maximus*, and *Manihot utilissima*, representing the main feedstocks for cattle in the study area. Mixed samples of 200 g were washed with 500 mL Milli-Q water before sealing in plastic bags.

### 2.3. Sample Preparation

#### 2.3.1. Soil Sample

Soil samples were oven-dried at 80 °C for 48 h, and their moisture contents determined [24]. Larger organic components were removed using a 2-mm sieve, and smaller components were removed by hand [25]. The samples were ground in a planetary micro mill and homogenized [26]. Subsamples of ~50 mg were mixed with a Pd-C and In internal standard (10 mg) and powdered using a mortar and pestle. A National Institute of Standards Rh solution (Wako Pure Chemical Industries, Osaka, Japan) was used as an internal standard. The Hg content of the US National Institute of Standards and Technology-certified soil reference material SRM 2782 was also determined for quality-control purposes.

#### 2.3.2. Forage-Plant Samples

Forage-plant samples were reduced in a blender before air drying for 24 h at 40 °C and powdering in a power mill (PM-2005m, Osaka Chemical Co., Ltd., Osaka, Japan). Samples of ~30 mg were mixed with 30 μL of a 10 ppm In internal standard solution and 1 mL 61% HNO_3_ before heating at 80 °C–100 °C for 2 min [27,28]. Samples were prepared for particle-induced X-ray emission (PIXE, Shimadzu Corporation, Kyoto, Japan) analysis by air-drying droplets of solution in a 4 μm-thick Prolene film (Chemplex Industries, INC., Florida, FL, USA).

### 2.4. Sample Analysis

PIXE is a useful ion-beam analysis technique (proton beams of 1–4 MeV energy) for the analysis of geological and biological samples. Soil and forage-plant samples were analyzed by PIXE at the Cyclotron Research Center, Iwate Medical University, Tomegamori, Takizawa, Iwate, Japan. Analytical accuracy was evaluated by analysis of the National Institute for Environmental Studies (NIES)-certified reference material No. 9 (Sargasso) with a certified concentration of Hg of 115 ± 9 ppm (dry weight). Calibration curves were produced using a multi-element calibration standard (Perkin Elmer, Massachusetts, MA, USA) [29]. The proton beam energy was 2.5–3.0 MeV, and a Si(Li) solid-state X-ray detector (Mirion Technologies, Munchen, Germany) was used, with X-ray attenuation by a 500 μm-thick Lumirrar (Polyester) plate (PANAC Corporation, Tokyo, Japan). A sensitivity of 200 pg was achieved with a 6-mm-diameter beam and a Ni foil and diffuser in a graphite collimator system, with a beam angle of ~35° to the horizontal axis [30]. 

### 2.5. Statistical Analysis

The Hg concentration data of Table 1 for the three soil and forage-plant sample sets are normally distributed, with similar variances. Analysis of variance (ANOVA) tests were applied to assess the significance of differences between Hg contents of soils and forage plants from the three sampling areas, and independent t-tests to assess differences between mining and control areas. *p* values of <0.05 were considered to indicate statistically significant differences. The software IBM SPSS Statistic 21 was used to produce summary statistics, test for normal distributions and homogeneity of data, and ANOVA.

## 3. Results

### 3.1. Hg Distribution in Soils

Soil mean; standard deviation; and median, minimum, and maximum Hg contents are given in Table 1. Twenty-six soil samples were analyzed, with the Hg content being highest in soils from the ASGM area (mean 390 ± 860 µg/g; *n* = 8), compared with control areas (mean 7.40 ± 9.90 µg/g; *n* = 6). Hg concentrations were also elevated in soils around commercial mining operations (13.0 ± 17.0 µg/g; *n* = 12). The maximum soil Hg content recorded in the ASGM area was 2500 ppm, and the lowest concentrations in the commercial mining and control areas were below the detection limit. Soil Hg contamination associated with trommel machine operations in ASGM areas was substantially higher than in other areas (Table 1).

An independent *t*-test showed no significant difference of Hg concentration in forage plants between each contaminated area with control area with (*p* = 0.30 and *p* = 0.47, respectively). One-Way ANOVA + Posthoc Tamhane’s showed there was no significant difference of Hg concentration in soil from three sampling area, with *p* > 0.05 (Table 1). The small sample size contributed no statistically significant difference in concentrations of Hg in soils between areas, even though the concentrations of Hg in soil were variable.

### 3.2. Forage Plants Hg Contamination

The Hg content of forage plants depends on environmental exposure, with concentrations varying with geographical location, soil type, groundwater, and ecological features. Mean, median, minimum, and maximum forage-plant Hg contents are given in Table 2. Forage-plant samples were analyzed from throughout the Bombana area, and results compared between the ASGM, commercial mining, and control areas. The mean Hg content was highest in forage plants from the ASGM area (9.90 µg/g compared with 2.70 µg/g in the control area). Hg contents were also elevated in forage plants in the commercial mining area (mean 3.20 µg/g). However, there was no significant difference in Hg contents of forage plants between contaminated and control areas (*p* = 0.33 and *p* = 0.79, respectively), and the ANOVA and Bonferroni posthoc tests indicate no significant difference between the three sampling areas (Table 2). This lack of difference might reflect the small number of samples.

## 4. Discussion

### 4.1. Source of Hg Pollution in the Terrestrial Environment

The Rarowatu areas are central to ASGM activity, which is undertaken by local communities in cooperation with outside investors [31]. Several stages of gold processing have the potential to produce Hg pollution. The major source of pollution is the ‘’roasting house’’, where gold ore is processed and where the trommel machine is housed. Ore grinding in the trommel machine and amalgam heating result in the release of Hg to the environment, with 50% of Hg being recovered and 50% lost (46% fused in tailings, and 4% evaporated to the atmosphere) [32]. Hg lost from ball mills and tailings also enters the environment and contaminates soil, water, and plants [33], while that released to the atmosphere is deposited downwind [9]. Trommel operators apply conventional methods of residue management to reduce the potential environmental impact. Tailings are stockpiled in underground areas or open holes to be used to build tailings dams. At some mine sites, dry tailings are transported to heaps or hillsides, or discarded in surface water [34].

Four commercial gold-mining operations, operating since 2008, are another potential source of Hg pollution (Figure 2). These mines exploit large-scale gold deposits and employ advanced and relatively clean technology, and heavy equipment. Their replacement of ASGM actually reduces Hg pollution because of their cleaner techniques. Land around commercial mining areas with potential commercial gold mining has been taken over by outside investors, although some illegal tailings processing occurs in commercial mining areas, where individuals use artisanal methods to extract the remaining gold. Illegal workers exposed to the amalgam-burning process will likely suffer health consequences, including kidney and central-nervous-system damage with tremors, insomnia, headaches, emotional changes, neuromuscular effects, and mental disorders [35,36]. 

### 4.2. Hg in Soils

The release of metals to the environment from the eight trommel houses in the ASGM region is the leading cause of soil Hg contamination. Although there are no trommel houses in commercial mining areas, various chemicals are used in gold purification.

The type of mining operation determines the level of metal contamination in soil, with waste release also affecting the form and degree of contamination (Figure 1). It is possible that elevated soil Hg levels in the control area are due to atmospheric deposition, as the control area is not far from the mining area (~6 km) [32,37,38]. 

Regulatory guidelines for soil metal contents include the Dutch Maximum Permissible Addition (MPA) for soils, which is used as a guideline for three hazard classes concerning vegetation, animals, and humans [39]. Concentrations above MPA values may pose serious risks. Comparison of soil Hg contents in the Bombana area with MPA guidelines indicates that contamination in the sampled ASGM area is at a highly hazardous level, with low–moderately hazardous levels occurring in the control area (Figure 1A). 

### 4.3. Hg in Forage Plants 

Mean Hg contents of forage plants are lower in the commercial mining areas than in the ASGM areas, with several gold-ore processing facilities contributing to the higher Hg levels. Forage plant samples from along the highway (the control group) have elevated Hg contents (Table 2). Forage-plant Hg contents were determined in the ASGM and commercial mining areas within 6 km of eight trommel houses and three gold-mining sites, with the control area being located along the highway with no mining activity nearby [9]. Critical limits for Hg in forage plants can be considered on the basis of its ecotoxicological effects on aquatic organisms, soil organisms, mammals, plants, and humans [39]. Such limits apply to several heavy metals affecting animal and human health, and causing phytotoxic effects in crops, with the latter being more stringent (Figure 1B) [40].

Hg contents of forage plants can be placed into three categories based on critical limits: high hazard (>3 ppm), low-moderate hazard (0.1–3.0 ppm), and low hazard (<0.1 ppm) [39]. Results indicate that Hg in forage plants in all sampled ASGM areas is at hazardous levels, and at low–moderate hazard levels in the control area.

### 4.4. Effects of Hg-Polluted Soil on Forage-Plant Growth

Plants require certain heavy metal for their growth and maintenance, in amounts that they can tolerate [41]. Excessive metal contents can be toxic to plants due to oxidative stress, inhibition of cytoplasmic enzymes, and cell damage [42]. Soil microorganisms also affect plant growth [43]. Previous studies [42,44] have demonstrated the effects of Hg on plant growth, biochemical structure, and physiology. For example, Hg contamination of tomatoes *(Lycopersicon esculentum)* reduces their rate of germination, stem height, fruit yield, and chlorosis. Likewise, in rice *(Oryza sativa)*, Hg contamination decreases tiller and panicle formation, resulting in a decrease in stem height and yield [42,45,46].

## 5. Conclusions 

Illegal mining in the artisanal and small-scale gold mining sector is the main source of anthropogenic environmental Hg emissions globally. Environmental impact results from discharge of Hg to the atmosphere, contaminating soil and vegetation. Bombana is a unique tropical savannah area with increasing ASGM activity. This study evaluated Hg concentrations in soil and forage plants, including ASGM and commercial mining, and control areas. 

The main finding of this study of environmental Hg contamination in the Bombana area is that Hg contamination of soil and forage plants is attributable to uncontrolled use of Hg in gold-ore processing. Concentrations of Hg in soil and forage plants generally exceed internationally accepted permissible guidelines. The anticipated growth of the ASGM sector will impose an environmental impact on inhabitants of ASGM areas; consequently, Hg pollution-reduction programs are critical in mitigating environmental hazards.

## Figures and Tables

**Figure 1 toxics-08-00015-f001:**
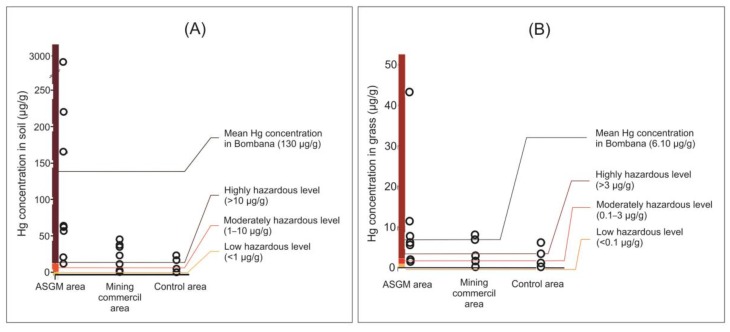
The regulatory guidelines and the hazard potential (**A**) distribution of soil sample and Hg level according to the MPA Dutch standards for soils and (**B**) distribution of forage plants Hg level according to the critical limits for Hg.

**Figure 2 toxics-08-00015-f002:**
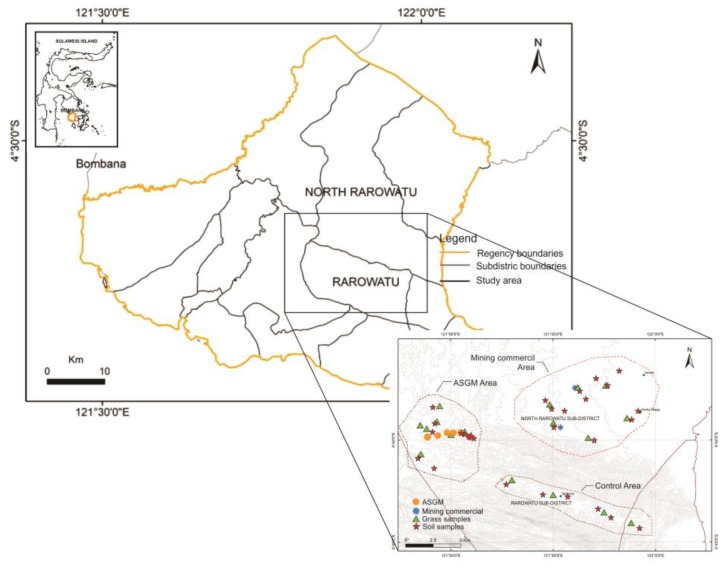
The locality of the study.

**Table 1 toxics-08-00015-t001:** Distribution of soil sample based on the minimum, median, mean, standard deviation, and maximum mercury (Hg) value.

Hg Concentration (µg/g)	Sampling Group		
	Control Area (*n* = 6)	Mining Commercil Area (*n* = 12)	ASGM Area (*n* = 8)
Minimum	0	0	12.0
Median	2.40	2.40	63.0
Mean ± SD	7.40 ± 9.90	13.0 ± 17.0	390 ± 860
Maximum	23.0	45.0	2500
Independent *t*-test (each contaminated area vs control area		(*p* = 0.30)**	(*p* = 0.47) **
One-Way ANOVA + Post hoc Tamhane’s			(*p* = 0.195) **

** = nonsignificant at *p* > 0.05.

**Table 2 toxics-08-00015-t002:** Distribution of forage plants sample based on the minimum, median, mean, standard deviation, and maximum Hg values.

Hg Concentration (µg/g)	Sampling Group		
	Control Area (*n* = 4)	Mining Commercil Area (*n* = 6)	ASGM Area (*n* = 8)
Minimum	0	0	1.50
Median	2.20	2.20	5.90
Mean ± SD	2.70 ± 2.80	3.20 ± 3.50	9.90 ± 14
Maximum	23.0	45.0	2500
Independent *t*-test (each contaminant area vs control area		(*p* = 0.33)**	(*p* = 0.79) **
One-Way ANOVA + Posthoc Bonferroni			(*p* = 0.354) **

** = nonsignificant at *p* > 0.05.

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
