# Peer review of "Mercury in Soil and Forage Plants from Artisanal and Small-Scale Gold Mining in the Bombana Area, Indonesia"

_toxics, 2020, doi:10.3390/toxics8010015_

Round 1

Reviewer 1 Report

I still have some comments to authors.

The Introduction section should be extended and autjors should add information about eg. Hg content in the grasses in different mining areas ect.

In the Materials and methods it should be add the subsection - Statistical analysis, where authors should add information about tests and softwere used in the article ect.

The Discussion part still needs to be improved, especially your results should be disscussed with results obtained by different authors (eg. Hg content in grass).

Author Response

January 13, 2020

Dear Reviewers of Toxics

On behalf of all the authors, I am pleased to resubmit for publication the revised version of toxics-690611 entitled “Mercury in soil and grass from artisanal and small-scale gold mining in the Bombana area” for publication in Toxics as an original research article. I appreciated the constructive criticisms of the Editor and the reviewers.

Here is my response related minor and major comment:

The Introduction section should be extended and authors should add information about e.g. Hg content in the grasses in different mining areas etc.

Response: I have extended the paragraph in the introduction section

Several researchers have investigated the concentration of mercury in plants that grows near various sources of mercury pollution including gold mines. The concentration of mercury in various samples of plants that grow around gold mines in various regions in Indonesia indicates the potential for environmental toxicity. The mine sites include Poboya in the city of Palu (6.5 ng/mg) [14], Sumalata District, North Gorontalo Regency (<0.00014-1.30822 mg/kg) [15], and Buru Island in Maluku (76.95 μg/kg) [16]. Meanwhile, analysis of plant samples collected from the Kaili coal mining region in China and from the Idrija mine area in Slovenia shows elevated of mercury (883 μg/kg and was 12,713 μg/kg, respectively) [17].

In the Materials and methods, it should be adding the subsection - Statistical analysis, where authors should add information about tests and software used in the article etc.

Response: I have adding the subsection for statistical analysis

2.4. Statistical analysis

The data sets from three sample group for soil and forage plants sample were normally distributed, and had the same variance. We used the analysis of variance (ANOVA) test to identify differences in Hg concentrations in soil and forage plants from three sampling area. Differences were considered significant for p values < 0.05. The statistical function in IBM SPSS Statistic 21 Ver. 21.0 were used for summary statistics, test for normal distribution and homogeneity of data, and analysis of variance (ANOVA).

The Discussion part still needs to be improved, especially your results should be discussed with results obtained by different authors (e.g. Hg content in grass).

Response: I have improved discussion section and added new subsection

Best Regards

Basri

Reviewer 2 Report

The updated version of this paper is much easier to read and understand.  There are still some areas that need clarification, as I have listed below.

Abstract:  include SD values with mean values.

Introduction:

P1, L40-42:  I think the “10-19 million workers” refers back to ASGM, so maybe move the sentence around?  “ASGM is now the main source of mercury pollution globally, with 10-19 millionn wokrers using mercury to purify gold in 70 countries.”  Then follow with the sentence that states “The amount of Hg released is proportional to 44 the volume consumed per unit of gold produced.” Then go back to “Mercury pollution also arises from non-ferrous metals, coal, the cement industry, and mine tailing waste.”

P2, L46-50:  Omit “in Bombana.”  “spread over large areas through wind and rain.”

M&M

P3, L76-77:  Adding the names of the grasses is really helpful.  You can use the scientific genus/species notation and italicize “Imperata cylindrica, Megathyrsus maximus, Manihot utilissima” but I think the last one is technically not a grass but a euphorb.  If this is the case, you need to replace “grasses” with “forage plants” throughout.

P3, L100-101:  This sentence structure is not correct.  “PIXE is an ion beam analysis (IBA) technique using protons with 1-4 meV of energy that is becoming an important tool for analyzing material, geologic, and biologic samples.” 

Results:

P4, L122:  “reached 2500 μg/g in one ASGM area”

Discussion: 

P5, L143:  “Natural processes that are sources of Hg in the atmosphere include volcano emitted Hg, at a rate between 80 and 600 tonnes/year.”

P5, L146-148:  This sentence needs to be referenced, especially since it could be controversial.  I was under the impression that most atmospheric mercury ends up in aquatic food chains.

P6, L159-160:  “tailings are transported to hillsides or dumped tailings into the surface air” this part of the sentence is unclear to me, please revise.

P6, L163-165:  “Some illegal tailings processing occurred in which individuals removed tailings and used artisanal methods to extract gold that was still present.”  I am also confused by the structure of this paragraph.  What land will be acquired?  The sentence about land doesn’t seem to refer to the preceding sentences in the paragraph.  Also, the last sentence seems to be taken directly from a reviewer question, but does not answer the question.

P6, L181-182:  Are you saying that the Hg detected in the control area is naturally occurring, or that it is caused by pollution from nearby mining areas?

P7, L197:  “A trammel house is a source of mercury contamination because it is where ore is ground with mercury in the amalgamation process, which produces small mercury droplets that are easily washed away and transported across large areas.”  I think this is what you want to say in this sentence.

Figure 4B:  The line that says “highly hazardous level (>3 μg/g)” points to approximately 45 μg/g on the graph.  Technically correct, but confusing, especially since the line below says “mean Hg concentration in Bombana (6.10 μg/g)” which is higher than 3 μg/g.

Author Response

January 13, 2020

Dear Reviewers of Toxics

On behalf of all the authors, I am pleased to resubmit for publication the revised version of toxics-690611 entitled “Mercury in soil and grass from artisanal and small-scale gold mining in the Bombana area” for publication in Toxics as an original research article. I appreciated the constructive criticisms of the Editor and the reviewers.

Here is my response related major comment:

Abstract:  include SD values with mean values

Response: I have included SD values with mean values in the abstract section

Abstract: Mercury (Hg) contamination in soil and forage plants is toxic to ecosystems, and artisanal and small-scale gold mining (ASGM) is the main source of such pollution in the Bombana area of Indonesia. Mercury contamination in soil and forage plants were investigated by particle-induced X-ray emission analysis of samples collected from three savannah areas. Forage plants Hg contents in the ASGM area (mean 9.90 ± 14 ppm) exceeded those in a control area (2.70 ± 14 ppm). Soil Hg contents (mean 390 ± 860 ppm) were also higher than those in the control area (mean 7.40 ± 9.90 ppm), with levels exceeding international regulatory limits. Hg contents in 69% of soil samples and 78% of forage plants samples exceeded critical toxicological limits, with contamination extending over large areas and posing a major environmental problem. Concentrations of Hg in soil and forage plants generally exceed internationally accepted permissible guidelines and being a major problem on miners and inhabitants of ASGM areas.

Introduction: P1, L40-42:  I think the “10-19 million workers” refers back to ASGM, so maybe move the sentence around?  “ASGM is now the main source of mercury pollution globally, with 10-19 million workers using mercury to purify gold in 70 countries.”  Then follow with the sentence that states “The amount of Hg released is proportional to 44 the volume consumed per unit of gold produced.” Then go back to “Mercury pollution also arises from non-ferrous metals, coal, the cement industry, and mine tailing waste.”

Response: As suggested by the reviewer I have revised the paragraph.

Natural source of Mercury (Hg) includes volcanic emissions, which release 80–600 tonnes Hg yr−1 to the atmosphere [7]. Geothermal activity, cinnabar deposits in geological formations, and forest fires contribute to the Hg atmosphere–deposition cycle. More than 90% of released Hg enters the terrestrial environment. ASGM is now the main source of mercury pollution globally [8], with 10-19 million workers using mercury to purify gold in 70 countries. The amount of Hg released is proportional to 44 the volume consumed per unit of gold produced. Mercury pollution also arises from non-ferrous metals, coal, the cement industry, and mine tailing waste (Fig. 2).

P2, L46-50:  Omit “in Bombana.”  “spread over large areas through wind and rain.”

Response: As suggested by the reviewer I have revised the paragraph.

A new gold-mining site was found in the Bombana area that has expanded progressively over the past ten years to provide potential mercury contamination of forage plants as animal feed. Gold processing activities can produce Hg particles that spread over large areas through wind and rain. Previous studies in the Bombana area indicated excessive dispersion of Hg, resulting in elevated Hg concentration in mining workers [6] and cattle [18]. There have been few studies of the effects of Hg pollution on soil and forage plants together, especially in savannah landscapes, with this being the first study of the spatial distribution of soil Hg contamination around ASGM sites and its effects on the forage plants to the best of our knowledge. This study aimed to evaluate the environmental implications of ASGM in the Bombana area, Southeast Sulawesi Province, Indonesia.

P3, L76-77:  Adding the names of the grasses is really helpful.  You can use the scientific genus/species notation and italicize “Imperata cylindrica, Megathyrsus maximus, Manihot utilissima” but I think the last one is technically not a grass but a euphorb.  If this is the case, you need to replace “grasses” with “forage plants” throughout.

Response: As suggested by the reviewer I have used the scientific genus/species notation and italicize “Imperata cylindrica, Megathyrsus maximus, Manihot utilissima”. I have replaced “grasses” with “forage plants” throughout as well.

Fresh forage plants samples were collected from 20 areas, including cattle-feed grasses (fodder) from various farms. Sample collections are preserved in the herbarium to provide evidence for vegetation for botanists to determine plant species [22]. At each sampling point, three main types of forage plants (Imperata cylindrica, Megathyrsus maximus, and Manihot utilissima) were drawn and mixed with weighing 200 grams. Samples were washed with 500 mL Milli-Q water before sealing in plastic bags. Three different species were sampled because vegetation is the main animal feed for cow colonies that live in the study area.

P3, L100-101:  This sentence structure is not correct.  “PIXE is an ion beam analysis (IBA) technique using protons with 1-4 meV of energy that is becoming an important tool for analyzing material, geologic, and biologic samples.” 

Response: As suggested by the reviewer I have change the sentence structure

PIXE is an ion beam analysis (IBA) technique using protons with 1-4 MeV energy that is becoming an important tool for analyzing material, geologic, and biologic samples [29].

P4, L122: “reached 2500 μg/g in one ASGM area”

Response: As suggested by the reviewer I have change the sentence structure

The maximum Hg concentration in soil samples reached 2500 μg/g in one ASGM area, while the lowest Hg concentration was 0.00 (non-detected) in mining company and control area (Fig. 2).

P5, L146-148:  This sentence needs to be referenced, especially since it could be controversial.  I was under the impression that most atmospheric mercury ends up in aquatic food chains.

Response: As suggested by the reviewer I have referenced the sentence

P6, L159-160: “tailings are transported to hillsides or dumped tailings into the surface air” this part of the sentence is unclear to me, please revise.

Response: As suggested by the reviewer I have revised the sentence

At some mine sites, dry tailings are transported onto heaps or hillsides or discarding tailings into the surface water.

P6, L163-165: “Some illegal tailings processing occurred in which individuals removed tailings and used artisanal methods to extract gold that was still present.”  I am also confused by the structure of this paragraph.  What land will be acquired?  The sentence about land doesn’t seem to refer to the preceding sentences in the paragraph.  Also, the last sentence seems to be taken directly from a reviewer question, but does not answer the question.

Response: As suggested by the reviewer I have revised the paragraph

The four commercial gold mining operations, which have been operating since 2008, are another potential source of Hg pollution (Fig. 1). Unlike the ASGM industry, the commercial sector manages large-scale gold deposits, using advanced technology and heavy equipment. The commercial operations are less polluting than the ASGM; therefore, wouldn’t bring more commercial operations actually decrease Hg pollution in the area by replacing ASGM operations with cleaner technology. However, some illegal tailings processing takes place in commercial gold mining areas, where individuals take tailings and use artisanal methods to extract the remaining gold. The Bombana region faces a future of severe Hg pollution. Currently, local inhabitant land located around commercial mining areas and indicating the economic potential for new commercial mines has finally been acquired and handed over by investors from outside Sulawesi.

P6, L181-182:  Are you saying that the Hg detected in the control area is naturally occurring, or that it is caused by pollution from nearby mining areas?

Response: As suggested by the reviewer I have answer the question

The operation of eight trommel houses in the ASGM area, all releasing heavy-metal waste into the environment, is the primary cause of high levels of Hg contamination there. Although there are no trommel houses in commercial mining areas, various chemicals are used in gold purification. Soil Hg levels are low in the control area caused by pollution from nearby mining activity with amalgam heating.

P7, L197: “A trammel house is a source of mercury contamination because it is where ore is ground with mercury in the amalgamation process, which produces small mercury droplets that are easily washed away and transported across large areas.”  I think this is what you want to say in this sentence.

Response: Ok, as suggested by the reviewer I have revised sentence to be better.

Mean Hg contents of forage plants samples were lower in commercial mining areas than in ASGM areas. Several sources of pollution at gold ore processing centers contribute to high Hg contents in forage plants. Forage plants samples from along the highway (the control group) had elevated Hg contents. Forage plants Hg contents were determined for two sites (ASGM and a commercial mining area) within 6 km of eight active trommel houses and three gold-mining sites. A trommel house is a source of mercury contamination because it is where ore is ground with mercury in the amalgamation process, which produce small mercury droplets that are easily washed away and transported across large areas [8]. The control area is located along the highway with no mining activity nearby. Critical limits for Hg in forage plants can be considered in three categories based on Hg-related ecotoxicological effects on aquatic organisms, soil organisms, mammals, plants, and humans [46]. These limits apply to several heavy metal including Hg as they affect animal and human health and have phytotoxic effects on crops [47]. Forage plants-quality criteria related to phytotoxic effects are more stringent than limits on animal and human exposure (Fig. 4B).

Figure 4B:  The line that says “highly hazardous level (>3 μg/g)” points to approximately 45 μg/g on the graph.  Technically correct, but confusing, especially since the line below says “mean Hg concentration in Bombana (6.10 μg/g)” which is higher than 3 μg/g.

Response: as suggested by the reviewer I have improved both of figure 4A and 4B.

Best Regards

Basri

Reviewer 3 Report

This is a very interesting article.  However, my main concern is the quality of the written paper and poor explanation of the results obtained and conclusions, particularly related to explanation of the findings. I am therefore reluctant to recommend publication of this article in Toxics without major changes. The authors should consider each section and review the content, particularly the results, discussion and conclusions. The methods, particularly the number of samples, is poor, making the paper weak; additionally, authors should consider performing a human risk assessment for the data found. The written English should be reviewed. Please consider my comments below:

Abstract

Please add the chemical symbol ‘Hg’ after mercury, so you can use it in the text. Please add conclusions.

I am not sure why different sections in the paper were highlighted in yellow, is there any reason for this?

INTRODUCTION

Line 37 – add “mercury” before the chemical symbol.

Line 53 – consider adding “to the best of our knowledge”.  

METHODS

Please consider highlighting the control area within Fig 1, and explain why that area was considered as control.

Why were the plastic bags used for the soil sampling autoclaved?

Please provide the scientific names for the grass samples in Italics. Please explain how species were identified for the grass samples, and why three different species were sampled. This may impact on the total concentration of Hg found, which authors should explain a bit further.

Please indicate the certified ref. materials that were used – be specific. Please provide recovery rates.

Authors should consider assessing human health risks to provide robustness to their paper. Authors should also explain and justify the low number of samples monitored.

I consider that the last sentence regarding the advantages of PIXE it is not of relevance for this paper or should be re-written.

RESULTS

This is one of the weakest sections of the article. There is lots of repetition, e.g. lines 117-121, lines 128-130.

Table 1 and 2. The statistics’ results are not very clear and p values and significance should be mentioned in the text.

Fig 2 and 3 – I consider that these do not provide much of value to the reader.

Authors provide very little results for a scientific article. Please consider assessing human risks and also adding more samples or information regarding other metals.

DISCUSSION

This is the weakest section of the paper. There is lots of information that should be placed in the introduction as it is not linked to the results obtained. For example, section 4.1. should be removed.

Section 4.1.1. – I consider that this paragraph is not appropriately justified. Please consider reviewing/revising it.

Section 4.1.2. – the first paragraph provides a sweeping statement as authors have not recorded or provided any atmospheric data. The first line is an introduction that should not be in the discussion section. Please avoid using the words “heavy metals” as this is not scientifically correct, use “metals” or “inorganic elements” instead.

Table 3 and Fig 4 – authors should consider merging them and provide this information in the results section? Lines 201-205 are not discussion. Lines 210-212 should be in the results section.

CONCLUSIONS

Conclusions should be re-written to make them clearer; currently some lines read like an abstract and authors provide statements that are too general/over extrapolated. In general, the conclusions are of little value to the scientific community.  

Author Response

January 13, 2020

Dear Reviewers of Toxics

On behalf of all the authors, I am pleased to resubmit for publication the revised version of toxics-690611 entitled “Mercury in soil and grass from artisanal and small-scale gold mining in the Bombana area” for publication in Toxics as an original research article. I appreciated the constructive criticisms of the Editor and the reviewers.

Here is my response related major comment:

Abstract: Please add the chemical symbol ‘Hg’ after mercury, so you can use it in the text. Please add conclusions.

Response: I have added chemical symbol “Hg” after mercury in abstract section, I add conclusions as well

Abstract: Mercury (Hg) contamination in soil and forage plants is toxic to ecosystems, and artisanal and small-scale gold mining (ASGM) is the main source of such pollution in the Bombana area of Indonesia. Mercury contamination in soil and forage plants were investigated by particle-induced X-ray emission analysis of samples collected from three savannah areas. Forage plants Hg contents in the ASGM area (mean 9.90 ± 14 ppm) exceeded those in a control area (2.70 ± 14 ppm). Soil Hg contents (mean 390 ± 860 ppm) were also higher than those in the control area (mean 7.40 ± 9.90 ppm), with levels exceeding international regulatory limits. Hg contents in 69% of soil samples and 78% of forage plants samples exceeded critical toxicological limits, with contamination extending over large areas and posing a major environmental problem. Concentrations of Hg in soil and forage plants generally exceed internationally accepted permissible guidelines and being a major problem on miners and inhabitants of ASGM areas.

Line 37 – add “mercury” before the chemical symbol

Response: I have added mercury before chemical symbol in introduction section.

Natural source of Mercury (Hg) includes volcanic emissions, which release 80–600 tonnes Hg yr−1 to the atmosphere [7].

Line 53 – consider adding “to the best of our knowledge”.  

Response: as suggested by the reviewer I have added “to the best of our knowledge”.  

There have been few studies of the effects of Hg pollution on soil and forage plants together, especially in savannah landscapes, with this being the first study to the best of our knowledge of the spatial distribution of soil Hg contamination around ASGM sites and its effects on the forage plants.

I am not sure why different sections in the paper were highlighted in yellow, is there any reason for this?

Response: paper highlighted in yellow indicates the sentence has been previously revised

Please consider highlighting the control area within Fig 1, and explain why that area was considered as control.

Response: as suggested by the reviewer I have highlighted the control area within figure 1 and explained why that area was considered as control.

Fresh forage plants samples were collected from 20 areas, including cattle-feed grasses (fodder) from various farms. Sample collections are preserved in the herbarium to provide evidence for vegetation for botanists to determine plant species [22]. At each sampling point, three main types of forage plants (Imperata cylindrica, Megathyrsus maximus, and Manihot utilissima) were drawn and mixed with weighing 200 grams. Samples were washed with 500 mL Milli-Q water before sealing in plastic bags. Three different species were sampled because vegetation is the main animal feed for cow colonies that live in the study area.

Why were the plastic bags used for the soil sampling autoclaved?

Response: The sample is stored in a plastic bag which is then transferred to an autoclavable polypropylene bag for pending analysis.

Please provide the scientific names for the grass samples in Italics. Please explain how species were identified for the grass samples, and why three different species were sampled. This may impact on the total concentration of Hg found, which authors should explain a bit further.

Response: as suggested by the reviewer

I have explained how species were identified for the grass samples I have provided scientific names for the grass samples in Italics I have explained why three different species were

Fresh forage plants samples were collected from 20 areas, including cattle-feed grasses (fodder) from various farms. Sample collections are preserved in the herbarium to provide evidence for vegetation for botanists to determine plant species [22]. At each sampling point, three main types of forage plants (Imperata cylindrica, Megathyrsus maximus, and Manihot utilissima) were drawn and mixed with weighing 200 grams. Samples were washed with 500 mL Milli-Q water before sealing in plastic bags. Three different species were sampled because vegetation is the main animal feed for cow colonies that live in the study area.

Please indicate the certified ref. materials that were used – be specific. Please provide recovery rates.

Response: as suggested by the reviewer I have indicate certified ref. materials that were used specifically but cannot provide recovery rates.

Soil samples were dried using oven at 80 °C over a period of 48 h accurate moisture content values for inorganic soils [23]. The larger organic parts of the material are removed using a 2-mm powder sieve, and the small parts are removed manually [24]. The soil samples ground in a planetary micro mill with three different durations; 5, 3 and 3 minutes (7, 7 and 9 speed, respectively), and homogenized [25]. Subsamples of ~50 mg were mixed with a Pd–C and indium internal standard (10 mg) and powdered with a mortar and pestle. The standard rhodium solution (Wako Pure Chemical Industries, Ltd., Osaka, Japan) was used as the internal standard and National Institute of Standard. Hg concentrations in soil were determined in the certified reference material National Institute of Standard and Technology (NIST) SRM 2782.

Authors should consider assessing human health risks to provide robustness to their paper. Authors should also explain and justify the low number of samples monitored.

Response: I do not measure human health risk because I have presented it in my previous published paper.

I consider that the last sentence regarding the advantages of PIXE it is not of relevance for this paper or should be re-written.

 Response: I prefer to remove this sentence because nor relevance for this paper

Table 1 and 2. The statistics’ results are not very clear and p values and significance should be mentioned in the text.

Response: as suggested by the reviewer I have revise my statistical test and mentioned in the text

Table 1. Distribution of soil sample based on the minimum, median, mean, standard deviation, and maximum Hg value.

Hg concentration (µg/g)

Sampling group

Total

Statistical Test

ASGM area

Mining commercil area

Control area

Number of cases

8

12

6

26

One-Way ANOVA + Post hoc Tamhane’s  (p = 0.195)**

Minimum

12.0

0

0

0

Median

63.0

2.40

2.40

14.0

Mean ± SD

390 ± 860

13.0 ± 17.0

7.40 ± 9.90

130 ± 490

Mean

390

13.0

7.40

130

Maximum

2500

45.0

23.0

2500

** = non-significant at p>0.05; * = significant at p<0.05.

Table 1 provided the distribution of soil sample based on mean, standard deviation, median, minimum and maximum Hg value.  A total of 26 soil samples were analyzed for Hg derived from various sampling points. The mean Hg concentration was highest in the soil from the ASGM area (390 μg/g) (control area: 7.40 μg/g, p <0.05). Hg concentrations were also higher than control in the soil of the mining company area (mean 13.0 μg/g). The maximum Hg concentration in soil samples reached 2500 μg/g in one ASGM area, while the lowest Hg concentration was 0.00 (non-detected) in mining company and control area (Fig. 2). There was no significant difference of Hg concentration in soil from three sampling area, with p > 0.05 (Table 1). 

Table 2. Distribution of forage plants sample based on the minimum, median, mean, standard deviation, and maximum Hg value.

Hg concentration (µg/g)

Sampling Group

Total

Statistical Test

ASGM area

Mining commercil area

Control area

Number of cases

8

6

4

18

One-Way ANOVA + Post hoc Bonferroni (p=0.354)**

Minimum

1.50

0

0

0

Median

5.90

2.20

2.20

3.20

Mean ± SD

9.90 ± 14

3.20 ± 3.50

2.70 ± 2.80

6.10 ± 9.80

Maximum

2500

45.0

23.0

2500

** = non-significant at p>0.05; * = significant at p<0.05

Section 4.1.1. – I consider that this paragraph is not appropriately justified. Please consider reviewing/revising it

Response: as suggested by I have moved the paragraph and merge into sub section 4.1.

Section 4.1.2. – the first paragraph provides a sweeping statement as authors have not recorded or provided any atmospheric data. The first line is an introduction that should not be in the discussion section. Please avoid using the words “heavy metals” as this is not scientifically correct, use “metals” or “inorganic elements” instead.

Response: as suggested by I have I have moved the first line to the introduction section and have change words “heavy metals” become metals and inorganic elements.

Table 3 and Fig 4 – authors should consider merging them and provide this information in the results section? Lines 201-205 are not discussion. Lines 210-212 should be in the results section.

Response: as suggested by I have merged table 3 to figure 4 because it contains similar information, but this portion retained in discussion section

Conclusions should be re-written to make them clearer; currently some lines read like an abstract and authors provide statements that are too general/over extrapolated. In general, the conclusions are of little value to the scientific community.

Response: as suggested by reviewer I have revised the conclusion

Round 2

Reviewer 1 Report

I don't have any other comments.

Author Response

February 3, 2020

Dear Reviewers of Toxics

On behalf of all the authors, I am pleased to resubmit for publication the revised version of toxics-690611 entitled “Mercury in soil and forage plants from artisanal and small-scale gold mining in the Bombana area, Indonesia” for publication in Toxics as an original research article. I appreciated the constructive criticisms of the Editor and the reviewers.

Thank you for your consideration!

Sincerely yours,
Basri
Sekolah Tinggi Ilmu Kesehatan Makassar
(Makassar Health Science Institute)
[email protected]

Reviewer 3 Report

The written English should be reviewed, especially in the newly inserted text/sentences. Please consider my comments below:

Line 24 - The last sentence of the abstract should be re-written.

Line 93 – please explain how forage plants species were identified and/or re-write the sentence. Do you mean that trained botanists identified the different species?

Why are the authors unable to provide recovery rates? Without them, we cannot know how reliable the experiment was, and therefore the results.

Authors indicated that they “do not measure human health risk because I have presented it in my previous published paper” – have you done this for Hg? Authors should briefly include this information in the paper and indicate the reference.

Table 1 – authors indicate that there are no statistical differences, but the concentrations of Hg in soils are very different, e.g. 390 ppm vs 7.40 ppm – maybe this is because of the very low number of samples monitored? Is there any differences between the different areas? Authors should explain this and consider using units instead of “ppm”.

Table 2 – please refer to my comments for table 1.

Line 196 – avoid using contractions

Line 200 – I do not understand the relevance of this sentence, authors should consider elaborating a little bit more.

Line 204 – This paragraph should be appropriately rewritten.

Line 243 – change the wording “heavy metals”. Authors use plural and singular terms throughout this paragraph, and there are words that seem to be incomplete such as “The of forage plants….” making it very difficult to follow.

Author Response

February 3, 2020

Dear Reviewers of Toxics

On behalf of all the authors, I am pleased to resubmit for publication the revised version of toxics-690611 entitled “Mercury in soil and forage plants from artisanal and small-scale gold mining in the Bombana area, Indonesia” for publication in Toxics as an original research article. I appreciated the constructive criticisms of the Editor and the reviewers.

Here is my response related major comment:

Abstract: Line 24 - The last sentence of the abstract should be re-written

Response: I have re-write the last sentence of the abstract        

Line 93 - please explain how forage plants species were identified and/or re-write the sentence. Do you mean that trained botanists identified the different species?

Response: I have re-write the sentence and make it simple and clear

Why are the authors unable to provide recovery rates? Without them, we cannot know how reliable the experiment was, and therefore the results.

Response: We have provide the certified concentration of Hg of 115 ± 9 ppm (dry weight)

Authors indicated that they “do not measure human health risk because I have presented it in my previous published paper” – have you done this for Hg? Authors should briefly include this information in the paper and indicate the reference

Response: We have provide the information of human health risk and included to the sentences

Table 1 and 2 – authors indicate that there are no statistical differences, but the concentrations of Hg in soils are very different, e.g. 390 ppm vs 7.40 ppm – maybe this is because of the very low number of samples monitored? Is there any differences between the different areas? Authors should explain this and consider using units instead of “ppm”

Response: I have explain in the paragraph ”why there are no statistical differences” and using units (µg/g).

Line 196 – avoid using contractions

Response: I have improved the paragraph become clear

Line 204 – This paragraph should be appropriately rewritten.

Response: I have re-written the paragraph become clear.

Line 243 – change the wording “heavy metals”. Authors use plural and singular terms throughout this paragraph, and there are words that seem to be incomplete such as “The of forage plants….” making it very difficult to follow

Response: I have change wording “heavy metals” and re-written the paragraph become clear.

Round 3

Reviewer 3 Report

Authors should consider collecting and analysing more samples to strengthen their paper. In the new communication, they should reflect their recoveries for the reference material used - this is critical to indicate that the methods used, specifically the PIXE, is able to determine Hg in the samples analysed. Authors only provide the concentration of Hg in the reference material used, this is not indicative of the percentage recovery.

This manuscript is a resubmission of an earlier submission. The following is a list of the peer review reports and author responses from that submission.

Round 1

Reviewer 1 Report

This article is very interesting and enlightening.  I hadn’t realized that Indonesia was such a major source of Hg pollution in the world. 

There are a few organizational problems, for example, the results includes some interpretation that belongs in the discussion section, and I think some of the conclusions reached in the discussion need to be better supported, for example, it is not clear how you reached the conclusion that the trommel equipment is a source of mercury.

It seems that figures 1 and 6 are very similar, do you need to have both of them?

Generally, I think it’s best to avoid the term “level” and instead use “concentration” since it’s a more specific, less vague term.  “Level” has other meanings. 

More specific comments are listed below:

Abstract: 

P1, L15:  Ecosystems and animals is kind of redundant, since animals are part of the ecosystem, but I’m not sure how to word this better.

P1, L17:  “Contamination levels” is kind of vague.  Maybe say “mercury concentrations in soil and grass . . .”

Introduction

P1, L39-41:  Can you split this into two sentences?  It starts out being about ASGM, then describes other sources of mercury, then goes back to ASGM.  It was difficult for me to follow and could be more clear if you made one sentence about the significance of ASGM in Hg pollution and a second sentence noting that there are other potential sources.

P1, L41-42:  The terms deposition and release have opposite meanings, so I suggest using the term “discharge” instead of “deposition.”

P2, L45:  Please explain what you mean by “biodegradation of Hg” here.  You can’t really degrade an element, you can only fix it in other forms (the way microorganisms produce methylmercury, for example). 

P2, L47-50:  Here you talk about natural sources of Hg.  It might be a good place to put the alternate anthropogenic sources that were in the confusing sentence L39-41.

P2, L53-54:  The food-chain sentence can probably be removed from here.  It mentions one of your conclusions (concentrations exceed international guidelines) and would work better in the discussion section.

M&M: 

P2, L61:  “commercial mining area.” Are these gold mining or other types of mines?  Can you give an idea of the scale?  Like, how many people are employed or how much income is generated?

P3, L74:  You note collecting “cattle-feed” grasses.  Do you mean you collected hay samples, or that you collected fodder (grass samples from the pasture) or were they just species of grasses that are commonly fed to cattle as hay and/or fodder?

P3, L82:  “Internal standard internal” can this be said in a way that is not so redundant?

P3, L96:  “PIXE is an ion beam analysis (IBA) technique used proton of 1-4 MeV energy and become an important tool. . . “  Are there words missing form this part of the sentence?  I’m having difficulty following it.

P3, L98:  “methods are: “ replace semi-colon with colon here.
P3, L100:  Sentence should end “with minimal sample preparation” because you already described drying/grinding/etc which counts as sample prep to me. You may also note if the method is sample-sparing (can you recover the sample and use it for further testing or whatever?)

Results:

P3, L102:  The heading isn’t clear.  “Mercury distribution in soil and concentrations found” or something like that?

P3, L106:  “with the lowest concentrations in the commercial mining and control areas.”  Concentrations (or levels) must be plural here because there is more than one location.

P3, L108:  Trommel machine is a new term for me.  Can you further explain in the materials and methods section how and why you sampled soil around trommel machines?  This will be helpful in supporting your statement that trommels are the major source of Hg in these areas.

P4, L115L  “Various source of natural processes" is unclear.  “Natural processes that are sources of Hg in the atmosphere include . . .” or something like that would work better.

P4, L116-121:  This paragraph does not fit in the results section. It should be rewritten for the discussion section or omitted entirely.

P4, L116-118:  I cannot follow this sentence. 

P4, L229-120:  This sentence has some problems with grammar.  Also, you appear to use land as an example of a terrestrial environments, but I think the terms are almost synonymous—I can’t think of any other examples of terrestrial environments besides land.

P4, L122:  The sentence needs to be restructured.  Total Hg was not derived from various sampling points.

P4, L124, P5, L141:  By elevated, you mean higher than control values?

P5, L133:  Remove the rest of this sentence starting with “indicating” because that belongs in the discussion section with further explanation (how does it indicate a source?)

P5, L135-137:  The first two sentences of this paragraph belong in the discussion section.

Discussion:

P7, L152:  Do you mean Figure 6?  Also, I can’t tell from the figure or the text where the new site was in 2008. Also, this figure seems to contain the same information as figure 1.

P7, L157:  Is the other 50% collected somehow?  So a total of 4% is actually volatilized into the atmosphere, and 46% is found in mine tailings.  Is there any attempt made to contain the tailings?  Please explain.

P7, L161:  “The four commercial gold mining operations, which have been operating since 2008, are another potential source of Hg pollution.”

P7, L163:  I don’t think you need to use the word “traditionally” here, it’s understood that commercial mining is different from artisanal mining.

P7, L164:  What is “illegal trailings processing”?  Does this mean individuals take the tailings and use artisanal methods similar to extract any gold that wasn’t already removed?  

P7, L164-165:  “The land will eventually be acquired” who owns the land now?  Is there potential for new commercial mines in the area? 

P7, L165-166;  base on your study, the commercial operations are less polluting than the ASGM, therefore, wouldn’t bringing in more commercial operations actually decrease Hg pollution in the area by replacing ASGM operations with cleaner techology?  Please explain this further in the discussion.

P8, L179:  “Comparison of soil Hg concentrations”

P8, L183: You need to support your statement that the trommel houses are the source of the contamination.

P8, L185:  Is there evidence of mining and amalgam activity in the control area, or are you speculating that they occur?  Is it possible that the Hg in the control area is naturally occurring or deposited from the atmosphere, since the control area is not far away from the mining areas?

P8, L187:  What do you mean by “local transportation”?  Are people moving tailings, or dust, or do you mean natural air and water currents?

Figure 2:  can you tell me what the “mercury sources” are?  You’ve already labelled the different mining areas and the control area, I would have expected the mining areas to be the “mercury sources.”  Also, why is Figure 2 not mentioned between Figure 1 and Figure 3 in the methods section?  Figures should be numbered as they appear in the sequence of the text.

Table 1:  Can you put the “minimum” values at the top under number of cases, and then the Median values?  I think it would be easier to be going from lowest to highest, and to keep the median and mean together for easy comparison, but you must also keep the mean and SD together.  Alternately, you can put the SD in parenthesis next to the mean.  “Mean (Std. Deviation) 390 ppm (860).   You need to include units on this table somehow.

Table 3:  I don’t think you need to include all of the other elements on this table if you are only concerned about Hg. 

References:  Is reference 1 a book?  There is no publisher listed.  I did not thoroughly review the references to make certain they meet the journal requirements.

Reviewer 2 Report

The aim of the article entitled “Mercury in soil and grass from artisanal and small-scale gold mining in the Bombana area, Indonesia” is to detect the Hg pollution in the area of artisanal and small-scale gold mining in the Bombana area. The problem discussed in the article is interesting and noteworthy, however, the text contains some errors/deficiencies requiring improvement, so in present form cannot be publish. Moreover, some parts of the text seem to be written carelessly. Please find some additional comments:

Introduction: should give some introduction of the problems discussed in the article, the background and generally this section is well written, however in my opinion here the authors should also give some information about eg. PIXE, so for this reason this section needs to be improve. Materials and Methods: should be improved

L.61 – Fig. 1, this is not a sampling location and I don’t see the sense of presenting Fig.1 and Fig.2. Both should be merged in one figure

67 – numeration of the figures (here should be 2 not 3).

L.74 -  did you determine the species of collected grasses?

75 – each type of grass? As above. 79 you cannot remove organic matter just using mesh, here you removed only larger organic parts 85 – something more about internal standard solution?

L.86 – on the basis of which procedure?

There is lack of information about statistical analysis.

Results:

Table 1 and 2 are not well prepared, should be constructed in different way

L.103 and 122 – the same text

Discussion part:

L.176 and 192 – there is the same text

L.203 and 212 sounds similarly

If we reduce those parts, the discussion part looks very poor, so for sure has to be improved, also in compare to the results of other authors in similar areas.

Conclusions: L 222 and 223 – I didn’t find any information in the text about the population density in this area, the land use and so on, so it should be also add to the text.